# Calving Localization at Helheim Glacier Using Multiple Local Seismic Stations

M. Jeffrey Mei[1,2], David M. Holland[3], Sridhar Anandakrishnan[4], and Tiantian Zheng[5]

[1]Department of Applied Ocean Science and Engineering, Woods Hole Oceanographic Institution, MA 02540, United States
[2]Department of Mechanical Engineering, Massachusetts Institute of Technology, MA 02139, United States
[3]Courant Institute of Mathematical Sciences, New York University, 251 Mercer Street, New York, NY 10012, United States
[4]Department of Geosciences, Pennsylvania State University, University Park, PA 16801, United States
[5]Department of Physics, New York University Abu Dhabi, PO Box 129188, Abu Dhabi, United Arab Emirates

*Correspondence to:* Jeffrey Mei (mjmei@mit.edu)

**Abstract.** A multiple-station technique for localizing glacier calving events is applied to Helheim Glacier in southeast Greenland. The difference in seismic-wave arrival times between each pairing of four local seismometers is used to generate a locus of possible event origins in the shape of a hyperbola. The intersection of the hyperbolas provides an estimate of the calving location. This method is used as the P- and S-waves are not distinguishable due to the proximity of the local seismometers to the event and the emergent nature of calving signals. We find that the seismic waves that arrive at the seismometers are dominated by surface (Rayleigh) waves. The surface-wave velocity for Helheim Glacier is estimated using a grid search with 11 calving events identified at Helheim from August 2014 to August 2015. From this, a catalogue of 11 calving locations is generated, showing that calving preferentially happens at the northern end of Helheim Glacier.

## 1 Introduction

The calving of marine-terminating grounded glaciers is a significant contributor to rising sea levels worldwide due to the massive volumes of ice involved that can suddenly be discharged into the sea. Depending on the glacier, the contribution of calving to sea-level rise can be equal to, or even greater than, the contribution from melt processes (Rignot et al., 2013; Depoorter et al., 2013). However, the lack of understanding of the physical principles that cause these events means that it is difficult to precisely forecast their contribution to sea level rise in the near future (e.g. Pfeffer et al., 2008; Meier et al., 2007). Calving glaciers can rapidly advance and retreat in response to minimal climate signals, which can rapidly change the sea level (Meier and Post, 1987; Nick et al., 2013). A better understanding of calving processes is vital to developing accurate predictions of sea-level rise.

The lack of understanding of why and how calving events happen makes it hard to create a general 'calving law' (Amundson and Truffer, 2010; Bassis, 2011). There have not been enough direct observations of smaller calving events (e.g. Qamar, 1988; Amundson et al., 2008) to identify patterns to attempt to form a general calving law. Calving events are intermittent, though they exhibit some seasonality due to the seasonality of the mélange ice, ocean temperature variations and variations in

basal motion due to meltwater input (Foga et al., 2014; Joughin et al., 2008). The overall unpredictability of calving requires monitoring equipment to be deployed on a long-term basis to detect events.

One way to monitor glaciers and detect calving is to use seismic arrays (e.g. Walter et al., 2013; Amundson et al., 2012; Köhler et al., 2015). Calving events can generate glacial earthquakes, with surface waves detectable at a teleseismic range (Nettles et al., 2008; Nettles and Ekström, 2010; Tsai et al., 2008). A common automated calving detection method is to use triggers based on the ratio of short-time-average and long-time-average seismic signals (STA/LTA). After an event has been detected, it can then be localized. Currently, most localization methods require visual confirmation of the calving location, unless they are sufficiently large to be seen by satellite imagery. Automatic methods like STA/LTA can help narrow down the manual search in satellite and camera imagery for calving but, ultimately, visually locating a calving event requires clear weather and well-lit conditions (O'Neel et al., 2007). An exception to this is terrestrial radar (e.g. Holland et al., 2016), but radar cannot be deployed year-round without constant refueling and swapping out of data drives, and also has problems seeing through atmospheric precipitation. Recently, high-frequency pressure meters, such as Sea-Bird Electronics tsunameters, have been deployed to monitor calving at Helheim (Vaňková and Holland, 2016).

Land-based seismometers offer improvements over simple camera or satellite imagery for detecting calving because seismic arrays are not limited to daylight hours, are not affected by snow, can be deployed year-round without maintenance and provide quantitative data to help estimate the magnitude of calving events. Seismic studies of calving have been done at the regional (<200 km) as well as the teleseismic level. Generally, teleseismic detections of calving are done via low-frequency surface waves (e.g. Walter et al., 2012; O'Neel and Pfeffer, 2007; Chen et al., 2011), while local detections are done at some subset of frequencies within 1-10 Hz (e.g. Bartholomaus et al., 2012; Amundson et al., 2008, 2012; O'Neel et al., 2007; Köhler et al., 2015).

Seismicity in glaciers has been observed for both basal processes (e.g. basal sliding) and surface processes (e.g. surface crevassing) unrelated to calving (Anandakrishnan and Bentley, 1993; West et al., 2010). Until recently, seismic signals generated by glacial calving were believed to be caused either by capsizing icebergs striking the fjord bottom (Amundson et al., 2012; Tsai et al., 2008) or interacting with the sea surface (Bartholomaus et al., 2012), or by sliding glaciers that speed up after calving (Tsai et al., 2008). Murray et al. (2015a) found that glacial earthquakes at Helheim Glacier are caused by glaciers temporarily moving backward and downward during a large calving event. Nettles and Ekström (2010) found that only capsizing icebergs generate observable low-frequency surface-wave energy, with calving events that create tabular icebergs not generating glacial earthquakes. Basal crevassing has also been suggested as a mechanism for calving at Helheim (Murray et al., 2015b). It is not yet known how to fully categorize and characterize different calving events.

Seismic signals of calving events typically have emergent onsets (i.e. having a gradual increase in amplitude with no clear initial onset) with dominating frequencies around the order of 1-10 Hz (e.g. Amundson et al., 2010; Richardson et al., 2010; O'Neel et al., 2007; Amundson et al., 2012). The emergent nature of the signals makes it hard to accurately identify a P-wave onset time, let alone a S-wave onset time, which hinders the traditional seismic triangulation method that takes the difference between the P- and S-wave arrival times to generate a distance to the epicenter (Spence, 1980). The other main method involves calculating backazimuths from a ratio of easting and northing amplitudes of P-waves from a broadband seismic station (e.g.

Jurkevics, 1988; Köhler et al., 2015); this fails for our study due to the proximity of our stations and the high speed of the sound waves (around 3.8 km/s through pure ice, e.g. Vogt et al. (2008)) which make the waves arrive near-simultaneously. Another method to locate calving events, known as beamforming, uses the seismic signals recorded on several array stations to determine the time delay associated with a backazimuth that aligns the signals coherently (Koubova, 2015). A more recent

method for localizing calving events is the use of frequency dispersion of surface waves, which uses a regional array (100-200 km away) of hydroacoustic stations to estimate a distance between event and detector and combines this with an azimuth (determined from the P-waves) to create a unique intersection (Li and Gavrilov, 2008), as the stations are sufficiently far to separate different seismic wave components. This method has similar precision to using intersecting azimuths from two remote stations, which is enough to identify at which glacier the calving occured, but not enough to localize the event within the

glacier.

In seismology, another technique to locate the epicenter of seismic events uses differences in signal arrival times to create a hyperbola, on which the epicenter lies. This was first used in Mohorovicic (1915); Pujol (2004) notes that this method is best for shallow events where refraction along a bottom interface (glacier-rock) is insignificant. Such a technique has not yet been applied to localizing calving. The aspect ratio (vertical/horizontal dimension) of Helheim Glacier is of order 0.1 and so

calving events should be sufficiently shallow to use this technique. This method is limited by determining the relevant wave velocity. In our case, this is empirically determined by using hyperparameter optimization, also known as grid search (Bergstra et al., 2013). This involves exhaustively evaluating a product space of parameters to optimize some performance metric. In our case, we use a product space of surface velocity $v_{\text{eff}}$, x-coordinate and y-coordinate, to minimize the total residual between the observed lags and the lag corresponding to each ($v_{\text{eff}}$, $x$, $y$). The hyperbolic method is then applied to calving events using

the mean $v_{\text{eff}}$ from the grid search, to localize the epicenters of the seismic signals generated during calving events. The grid search is then repeated with a product space of just ($x, y$) with the mean $v_{\text{eff}}$ from the first grid search, and these localizations are compared to the hyperbolas.

## 2   Data

Four broadband seismometers (HEL1: Nanometrics Trillium 120, HEL2-HEL4: Nanometrics Trillium 240) with sampling

rates of 40 Hz and 200 Hz were deployed around the mouth of Helheim Glacier (Figure 1). HEL1 and HEL2 were deployed in August 2013, while HEL3 and HEL4 were deployed in August 2014. They were synchronized with Coordinated Universal Time. These stations detected seismic activity from both calving as well as distant earthquakes, so we first inspect the frequency distributions of the signals to isolate calving events.

A calving event that was observed *in situ* at Helheim in August 2014 (Figure 2a) matches those of O'Neel et al. (2007),

Richardson et al. (2010), and Amundson et al. (2012) very well in both frequency distribution and shape, with an emergent onset and relatively high-frequency signals (1-20 Hz). In contrast, events from regional earthquakes have much lower-frequency signals (<1 Hz). A M5.2 regional earthquake in Bárðarbunga, Iceland on September 1 2014[1] (Figure 2b) shows that the

---

[1]Icelandic Meteorological Office record: http://en.vedur.is/earthquakes-and-volcanism/articles/nr/2947

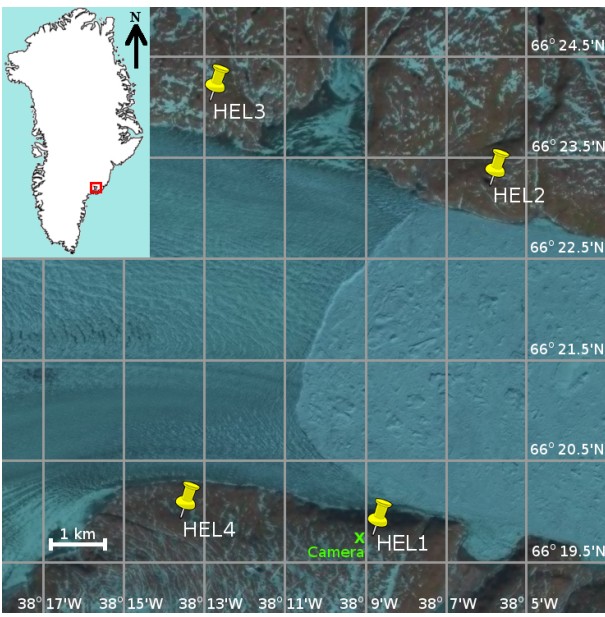

**Figure 1.** The four bedrock-deployed seismometers deployed at Helheim Glacier as shown on a Landsat-8 image from July 9 2015. GPS coordinates are referenced to WGS84. HEL1: 66°19.76′N 38°8.79′W. HEL2: 66°23.24′N 38°5.91′W. HEL3: 66°24.06′N 38°12.9′W. HEL4: 66°19.94′N 38°13.60′W. The calving front is clearly visible in between them. Westward is Helheim Glacier; eastward is the mélange and Sermilik Fjord. A camera was also set up next to HEL1.

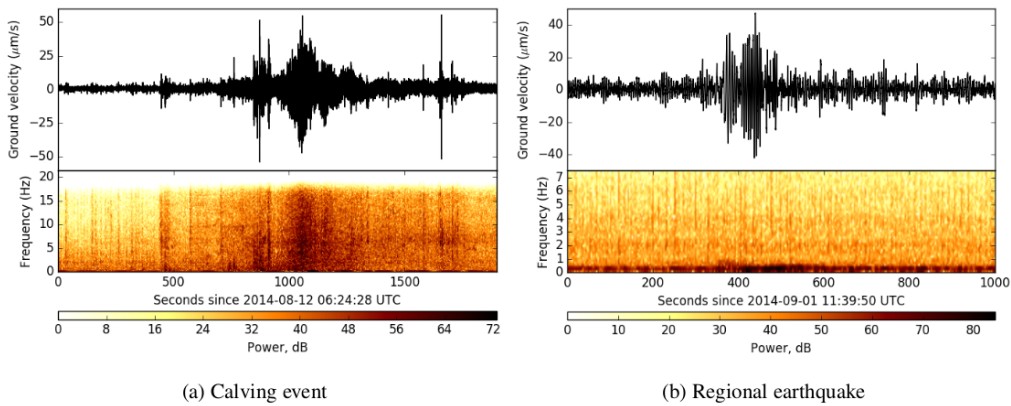

(a) Calving event          (b) Regional earthquake

**Figure 2.** Spectrograms for (a) a calving event at Helheim on August 12 2014, and (b) a regional earthquake in Bárðarbunga, Iceland on September 1 2014. The easting amplitude of the seismometers is used for both events. The seismogram (top) and spectrogram (bottom) of each event share the same time axis for direct comparison. The spectrograms have a window size of 256 points (= 6.4 s).

dominant frequencies received at the HEL seismometers are all well below 1 Hz. This means we can easily separate calving events from regional seismic activity by using a bandpass filter (Butterworth, two-pole and zero-phased). We bandpass-filter

between 2-18 Hz based off the spectrogram in Figure 2a in order to maximize the signal-to-noise ratio. Using some threshold of STA/LTA counts, we are able to create a catalogue of 11 calving events on which to run our hyperbolic method algorithm. This ignores smaller calving events, which generally have amplitudes too small to easily identify a signal onset. Calving events, with the exception of events in January/February 2015 for which imagery is too snow-covered to use, are confirmed with local
camera imagery and MODIS satellite imagery from the Rapid Ice Sheet Change Observatory (RISCO)[2].

## 3  Localization methods and results

### 3.1  Hyperbolic method

After isolating the calving events, we apply the hyperbolic method to generate a catalogue of calving locations. A hyperbola can be geometrically defined as the locus (set of points) with a constant path-difference relative to two foci, as seen in Figure 3.
In our case, each pair of seismometers acts as foci. We need two variables to determine the path difference: the signal-arrival time lag at each pair of seismometers, and the horizontal velocity of the surface waves.

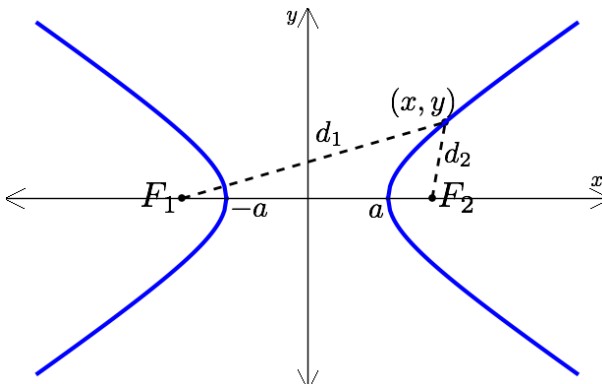

**Figure 3.** An example of a hyperbola of equation $x^2/a^2 - y^2/b^2 = 1$, with foci at $F_1$ and $F_2$ with constant path difference $|d_2 - d_1| = 2a$. $b$ can be generated by $\sqrt{c^2 - a^2}$, where $2c$ is the known distance between the foci.

Assuming that the speed of seismic waves across Helheim does not vary horizontally, the signals from a calving event that happened exactly at the midpoint of the two seismometers (or any other point along the perpendicular bisector of the two seismometers) would arrive simultaneously at the two seismometers. Similarly, if the event happened closer to HEL1, the
seismic waves would arrive slightly earlier to HEL1, and the locus of possible calving locations would instead be the set of all points whose distance from HEL1 is shorter than HEL2 by a fixed length. This length is $2a$ (Figure 3), which is the product of the speed of the waves through the glacier ($v_{\text{seismic}}$) and the time lag in signal arrival ($\Delta t$) and is defined for a hyperbola with equation $x^2/a^2 - y^2/b^2 = 1$. We may use the time lag of the signal arrivals at the two seismometers (which become the foci) to determine the path difference of the signals to form the locus. One of the curves (either the left or right in Figure 3)

---

[2]http://www.rapidice.org/viewer/

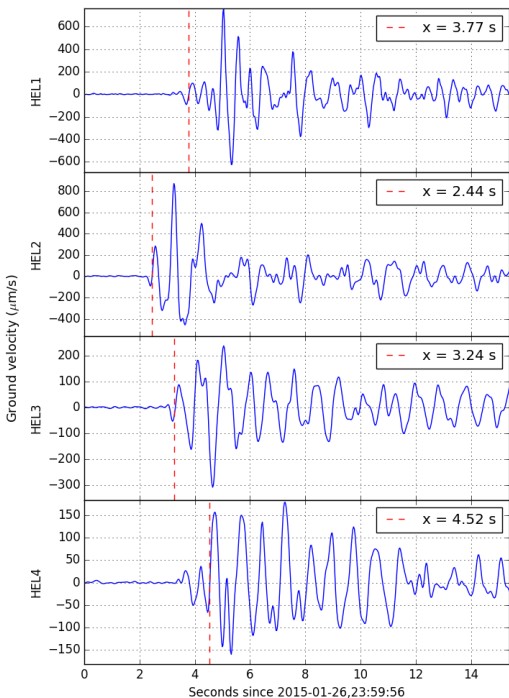

**Figure 4.** Seismic signals for a calving event at Helheim Glacier on January 26 2015. The signal onset times are determined using an automated script that searches for the first instance of a gradient exceeding a particular threshold as defined in Section 3.2. The differences in the wave onset times are then used to generate a characteristic path difference for each hyperbola.

may always be eliminated as we know which seismometer the event occurred more closely. Each time lag therefore generates one curve that intersects uniquely with the calving front, which will give the location of the calving. If the calving front is not known, the calving event can be triangulated using additional pairings of other stations.

This method requires evaluating the time lag between the signal arrival times at each seismometer (Figure 4), and obtaining the speed of the seismic waves through the glacier. As the surface waves travel over a topography unique to each glacier, we rename the variable as $v_{\text{eff}}$, which is the effective speed of the seismic packet over the surface of Helheim Glacier using the above assumptions.

## 3.2 Identifying signal lags

To identify the time lag, we first try using cross-correlation of the signals. For subpanels HEL2 and HEL4 in Figure 4, cross-correlation gives 1.5 s, which is a plausible value by eye, but for subpanels HEL3 and HEL4, cross-correlation gives 2.2 s which is not plausible by eye. The signals in Figure 4 do look qualitatively different for HEL3 and HEL4, and it is possible that this is what prevents cross-correlation from generating an accurate lag time. Instead of using cross-correlation, we use an automated script that searches through the signal for the first instance of a raw waveform gradient exceeding 1.44 standard

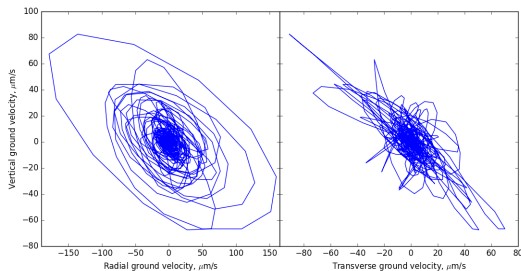

**Figure 5.** Particle plots of seismic wave arrivals for the calving event of July 7 2015, split into radial and transverse components. The characteristic elliptic shape of the surface Rayleigh wave is clearly visible in the radial component of the particle plot.

deviations of all point-wise gradients at each time step of 0.025 s for the total time window in Figure 4. This value of 1.44 was empirically determined as this produced the closest match to cross-correlation for signals that were qualitatively similar enough to use cross-correlation.

### 3.3 Determining seismic wave velocity with grid search

From particle motion plots (Figure 5), we know these signals are dominated by surface waves. We assume that the seismic wave travels at the same lateral speed from the calving epicenter to each station. The dependence of wave speed on glacier depth is not important for this method as long as the effective (surface) lateral speed to each seismometer is the same in each direction. We also assume that the glacier surface, calving epicenter and seismometers are all coplanar, so that the hyperbolas can be kept two-dimensional for simplicity. In reality, there is some elevation between the seismometers and the glacier surface, though

this distance (<300 m) is so much shorter than the seismometer separation (>6000 m) that refraction at the ice/rock boundary is likely negligible for characterizing the hyperbola. However, this method would become more precise with three-dimensional hyperboloids instead of two-dimensional hyperbolas.

We apply a grid search (hyperparameter optimization) to find the optimal $(x, y, v_{\text{eff}})$ to minimize the sum of the residuals of the time lags that would occur at each $(x, y)$ for that $v_{\text{eff}}$ as compared to the real observed time lags at each station. We

parametrize between $1.00 < v_{\text{eff}} < 1.40$ km/s (step size 0.01 km/s) and the coordinate span of the entire map in Figure 1 (step size 1 pixel) for our 11 identified calving events, and get a mean $v_{\text{eff}} = 1.20$ km/s with a standard deviation $\sigma = 0.1$ km/s. The standard error for these 11 samples is therefore $\sigma/\sqrt{11} = 0.03$ km/s. For all further plots, we therefore use $v_{\text{eff}} = 1.20$ km/s. We generate four hyperbolas, using HEL1-HEL2, HEL1-HEL3, HEL2-HEL4 and HEL3-HEL4 as these have the greatest distance of ice between the stations, as we require that the rock has a negligible contribution to the wave arrival times.

### 3.4 Localization results

Once we generate four hyperbolas we may take their intersection to be an estimate of where the calving occured. In Figure 6, we show the progression of one calving event on June 6 2015. From this, the main peak (blue) corresponding to the highest

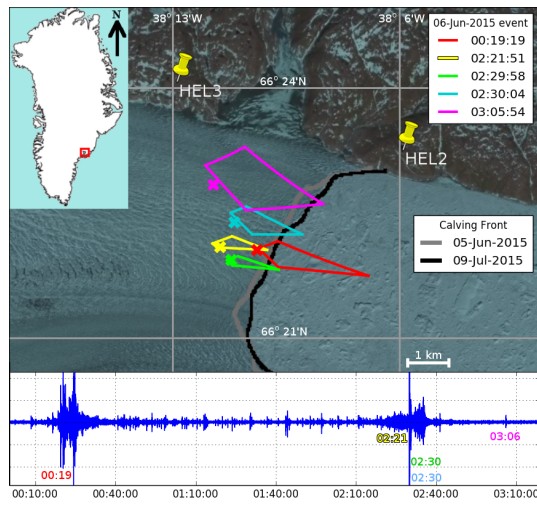

**Figure 6.** The calving event from June 6 2015, with the localizations (top panel) and the easting amplitudes of seismometer HEL1 (bottom panel) showing several sub-events. The x's indicate locations derived from using a grid search through a lattice of all points on the map with a fixed $v_{\text{eff}}$ = 1.20 km/s.

amplitude signal is taken as a representative location for the entire event for the purposes of creating a catalogue of all events from August 2014-August 2015. Applying this method to our entire catalogue of 11 calving events yields Figure 7. We also re-run our grid-search method, this time with a fixed $v_{\text{eff}}$ = 1.20, as a check of our localization results.

## 4  Discussion

### 4.1  Interpretation of Results

The hyperbolic method and grid-search method give very similar localizations for calving events at Helheim. Qualitatively, Figure 6 shows that calving propagates up-glacier, with an initial event near the calving front (red) and subsequent seismic signals originating from locations further up the glacier. The locations of events also diverge, as after the second event (yellow), the third and fourth events (green and blue) go in opposing directions. Given that the calving front depicted in grey corresponds to one day before the calving event, the fact that the first event (red) is localized so close to the calving front is a good indicator that the event is localized correctly. Similarly, the year-long catalogue in Figure 7 has events being localized near the calving front. For example, the black event of July 7 2015 is localized, for both the hyperbolic method and grid-search method, immediately adjacent to the black calving front corresponding to July 9 2015. Moreover, local camera imagery (Fig. 8) also shows substantial ice loss on July 7 2015 on the southern half of Helheim Glacier. We are therefore confident that the hyperbolic method and grid search method are valid methods to localize calving.

Based on Figure 7, calving appears to cluster in the northern portion of Helheim Glacier. This is consistent with the topography of the bedrock at Helheim (Figure 9), where the northern half is on the order of ∼200 m deeper than the southern half

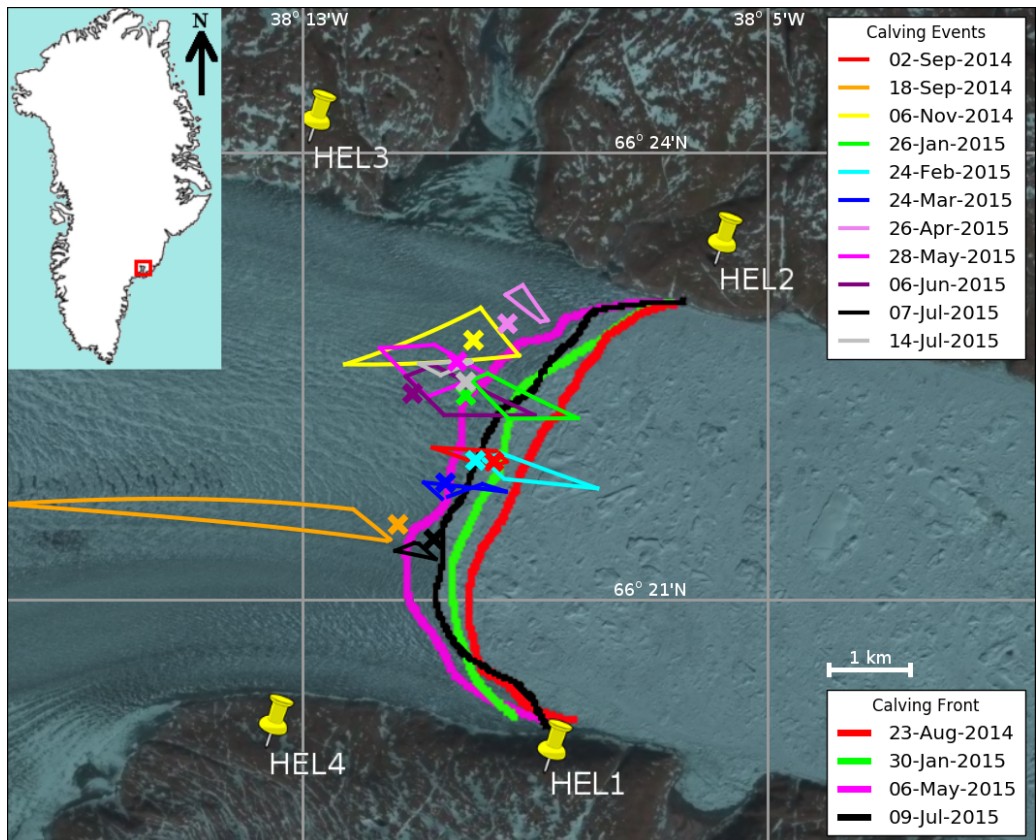

**Figure 7.** Catalogue of all calving events with clear signal onsets at Helheim Glacier from August 2014-August 2015 overlaid on Landsat-8 imagery of Helheim Glacier. Each color corresponds to a calving event, with only the area of overlap of the four hyperbolas being depicted. The x's represent the same event located using a grid search technique.

(Leuschen and Allen, 2013). It is possible that the deeper the ice, the higher the freeboard of the ice front and the greater the stresses that affect the calving front. In Figure 7, we see wider gaps between crevasses in the north of the glacier as compared to the south. This may also mean that the surface velocities are different in each half, which would affect the localization results. The topographic differences of both the glacier surface and ice bottom may contribute to why we see calving primarily in the northern half of Helheim.

It is possible to constrain the fault size of the rupture caused by calving. Using a shear model from Brune (1970), the radius $r_0$ of a circular fault is inversely proportional to the corner frequency $f_c$ of a S-wave and is given by

$$r_0 = \frac{K_c \beta_0}{2\pi f_c}$$

where $\beta_0$ is the shear velocity and $K_c$ is a constant, equal to 2.34 for Brune's source model (Gibowicz and Kijko, 2013). From Figure 10, the corner frequency is approximately bounded between 5 and 10 Hz. Taking a Poisson ratio of 0.3 for ice (Vaughan, 1995), the ratio of the Rayleigh-wave velocity to S-wave velocity is approximately 0.930 (Viktorov, 1970), giving

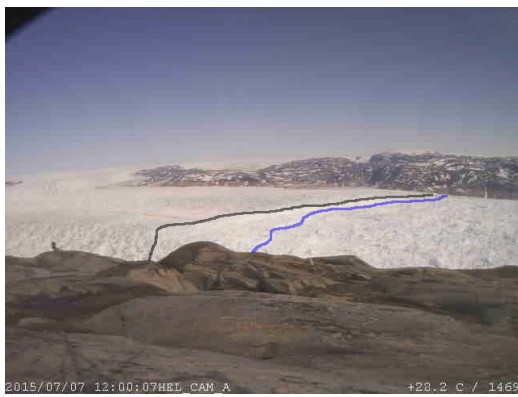

**Figure 8.** Local camera imagery for the calving event from 07-Jul-2015. The blue line indicates the calving front from the last image taken before the calving event, and the black line indicates the first image taken after the calving events. Images are taken every hour. The position of the camera is given in Fig. 1

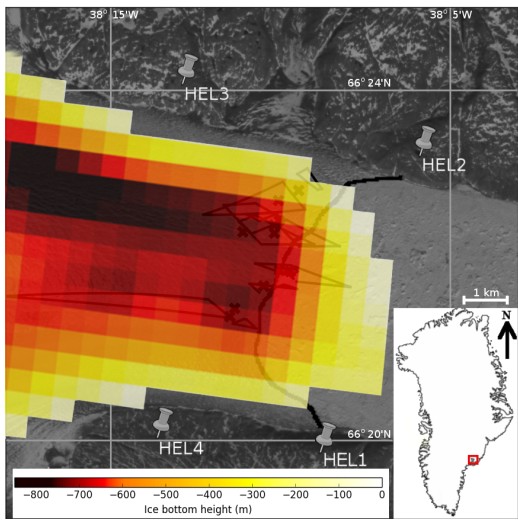

**Figure 9.** The calving events from Figure 7 overlain with the bedrock topography from the Multichannel Coherent Radar Depth Sounder (MCoRDS) L3 data set from NSIDC (Leuschen and Allen, 2013), with the calving front from July 09 2015 in black. The topography is collated and averaged from 2008 to 2012.

a value of $\beta_0$=1.29 km/s. For this rough calculation, we assume that the corner frequency is the same for the Rayleigh and S waves. This bounds the fracture size of the calving event between $48$ m and $96$ m. Brune's relationship does not depend on properties of the material like effective stress $\sigma$ or rigidity $\mu$. Our range of $48$ - $96$ m is considerably smaller than a typical observed calving fracture by around one order of magnitude. A fracture size of order 1 km would require a corner frequency of order 0.1 - 1 Hz, which we do not observe. 100 m is more on the order of a crevassing event, which also occur during/before events, so it is possible that crevassing events continue to happen during the calving event and obscure the power spectrum

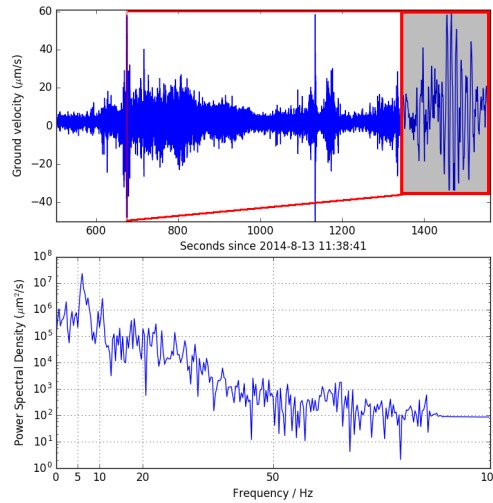

**Figure 10.** A typical power spectrum for a calving event (August 13 2014), for a 3-second time window containing the highest peak amplitude of the event. The shaded inset in the top panel shows a zoomed-in view of this window.

seen in Figure 10. Both basal crevassing (e.g. Murray et al., 2015b; James et al., 2014) and surface crevassing (e.g. Benn et al., 2007) have been suggested as calving mechanisms. Basal crevassing may be a more plausible explanation for Helheim, as Murray et al. (2015b) found that buoyant flexure via basal crevasses was the dominant cause for calving at Helheim in 2013. Our estimated rupture sizes using Brune's model could plausibly be the size of either, and as our method assumes a planar
glacier surface, we cannot distinguish whether the crevassing is at the base or the surface.

### 4.2    Discussion of methods

The hyperbolic method described in this paper offers some benefits to traditional seismic location techniques, which are more suited for regional seismic arrays that can distinguish between the different seismic wave types (e.g. O'Neel et al., 2007). Moreover, regional arrays do not give the kind of precision that local arrays would have, as small errors on a regional azimuth
translate to a large area of uncertainty on the local glacier surface. The hyperbolic method takes advantage of the stations' proximity to calving events and does not require separating out the different wave phases, thus sidestepping the P-wave identification problem that hampered localization techniques from Amundson et al. (2008) and Richardson et al. (2010).

The method also offers advantages over traditional calving detection methods, which require the use of a local camera and/or satellite data to visually confirm that calving took place. As seen in Amundson et al. (2012, 2010), calving generates
a characteristic seismic signal (Figure 2) that is easily distinguishable from signals from regional earthquakes. This is likely because higher frequency signals from regional earthquakes are attenuated by the time they reach the seismometers. This allows seismometers to be used to monitor glaciers and quickly identify calving when power in the 2-18 Hz range exceeds

some ratio above the ambient noise. Importantly, this monitoring could take place year-round, during the night and also cloudy days, making it a helpful addition to locating calving alongside satellite imagery, camera imagery and radar monitoring.

The seismic signals detected during calving events are clearly dominated by surface waves. Particle plots (Figure 5) show the characteristic elliptical shape of a Rayleigh wave. The Rayleigh waves, which are in theory parallel to the vertical axis, appear
slanted in Figure 5. It is possible that the mix of different wave phases (e.g. Love waves, also a surface wave) has interfered the Rayleigh wave such that it is no longer parallel to the vertical axis. There is also a lack of linear polarization as would be expected for a P-wave. Our estimated S-wave velocity, using a Poisson ratio of 0.3, is 1.29 km/s from above. This is lower than the 1.9 km/s for S-waves in pure ice that Kohnen (1974) found. It is possible that this is due to the anisotropy of the glacier surface, such that the ice is cracked and the seismic waves do not travel through pure ice. Given our characteristic surface
wave velocity on the order of 1 km/s with frequencies of order 10 Hz (see Figure 2), this corresponds to a surface wavelength of order 100 m. This is small enough to be affected by crevasses along the surface of the glacier which are of similar depths (Bassis, 2011). This means that we can reasonably expect these crevasses to affect the seismic wave velocity, which could slow the S-waves and surface waves, making our surface wave speed of 1.20 km/s a plausible value.

Because we are only working with surface waves, this limits our localization technique to just the epicenter of a calving
event, with no suggestion of a focal depth. This means we could not distinguish between basal or surface crevassing, even if we could estimate a rupture size in the previous section. Moreover, we have assumed a planar ice front for simplicity. It is possible that this method could be extended to determine the depth at which calving (or crevassing) occurs by using a 3-D hyperboloid instead of 2-D hyperbolas.

The calculation method we have used ignores the presence of the rock between the glacier and the seismometers, as the
proximity of the seismometers to the glacier means that the time taken for the wave to propagate through rock is negligible. Our method does not take into account the refraction at the ice-rock interface. Due to the ice dominating the wave path from the source to the seismometers, we assume that the refraction has a negligible affect on the trajectory of the surface waves.

The main source of error comes from identifying the signal onset. Picking out the signal onset is not fully automated because it requires setting a gradient threshold manually, or manual checking the plausibility of cross-correlation results. Local stations
that are right by the calving front are subject to much more noise than regional arrays. While some of the noise can be filtered out, a lot of the noise still occurs in the 2-18 Hz range that also contains most of the power from the calving signal. Moreover, as the calving events occur between the stations, the signals that arrive at each station come from different directions and may not necessarily be similar in shape. As a result, cross-correlation does not always work for determining lags. We cannot cross-correlate the envelopes as this would lose resolution of the lags (the envelope is of order 5 s in Fig. 4 but we have lags
of order 1-3 s and even a 0.5 s shift would dramatically change the hyperbola). Our empirical method of using gradients is not rigorous as it requires manual confirmation; this means the error is difficult to quantify as the true signal onset time is not known. However, the $v_{\text{eff}}$ of the surface waves can be estimated using a grid search method, giving plausible results. With more calving detections, the standard error of the optimized $v_{\text{eff}}$ value will decrease. As cross-correlation does work for some events, with a sufficiently large number of calving events, we may simply discard events that do not cross-correlate correctly. This
would make it possible to create an event catalogue using only automated methods.

## 5 Conclusions

Our results show that calving can be localized with local seismic stations. We find that the local seismic signals are dominated by surface waves, and that the differences between these signal onsets can be used to localize calving. This offers an alternative to regional arrays, which can distinguish different wave phases but have lower resolution of localization. Identifying the signal onsets can be automated, but still requires manual confirmation of results. Further study should be done in determining why cross-correlation only works for a subset of the events. With three or more seismometers, calving events can be detected and triangulated even without any satellite or camera imagery. Our catalogue of calving events at Helheim suggests that in the 2014-2015 season, calving typically initiated at the northern half of the calving front, which will help to constrain model simulations of glacier dynamics at Helheim. This technique can be applied to localize calving events at other glaciers.

*Acknowledgements.* The fieldwork necessary to collect this seismic data was made possible by the Center for Global Sea Level Change, grant G1204 of the NYU Abu Dhabi Research Institute and the Undergraduate Research Fund at NYU Abu Dhabi. Denise Holland provided the field logistics support in Greenland. The authors also acknowledge the support of the Arctic Division of the Office of Polar Programs under grants ARC-0806393 and ARC-1304137. Technical installations were performed by Paul Carpenter and Jason Hebert of PASSCAL Instrument Center. The authors would like to thank Jason Amundson, Anja Diez and an anonymous reviewer for their helpful comments in improving the manuscript.

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
