# Peer review of "Calving Localization at Helheim Glacier Using Multiple Local Seismic Stations"

_The Cryosphere, 2016_

## Referee Comment (RC1) · Anonymous Referee #1 · 3 Jun 2016

The authors apply a method to localize seismic calving events recorded close to the glacier terminus. Traditional travel-time based localization methods often fail close to the calving front since signal waveform are too complicated to identify distinct seismic phases. The method here makes use of arrival time differences of a calving signal between two stations which can be used to construct a hyperbola which either intersects with the know position of the glacier terminus or with other hyperbolas if more than two stations are available.

First of all, I would to thank the authors for their contribution to the progress in the emergent field of cryo-seismology. Studies like this can help to improve and establish passive seismic methods for monitoring and better understanding glacier dynamics.

(1) My main comment is that the method is actually not new in seismology or acoustics,

but has been suggested as a simple graphical signal localization method before, e.g. in:

J. Pujol, Earthquake location tutorial: graphical approach and approximate epicentral location techniques, Seismol. Res. Lett., 75 (2004), pp. 63–74

M. Bath, "Introduction to Seismology", (2nd ed.), Birkhäuser Verlag, Basel, Switzerland (1979)

M. Dragoni, , M. Gasperini, "On the localization of seismic events", Article, La Rivista del Nuovo Cimento (1978-1999), Volume 5, Issue 11, pp 1-28

I. Awaludin, A. S. Prihatmanto, E. M. I. Hidayat and C. Machbub, "Hyperbola tracing algorithm based on particle filter approach within a half-quadrant space for signal source localization," 2015 5th IEEE International Conference on System Engineering and Technology (ICSET), Shah Alam, 2015, pp. 17-22. doi: 10.1109/IC-SEngT.2015.7412438

I agree that this study present the first application of this method to calving localization. However, this is not such a different approach compared to using traditional localization methods based on first onsets (without using S waves). Basic processing and assumptions are the same: you have to know the velocity model and to pick first arrival. Just using P wave onsets, it is necessary to use some constrains on source locations, but you do the same here when choosing one hyperbola.

The authors should clearly write this and state some references.

(2) The title suggests a two-station method. This is true if the location of the terminus is known. However, in the paper the authors only present results of using more than two stations at Hellheim and Jakobshavn glacier (hyperbola intersection). So I would suggest to change the title or put more emphasize on two-station results in the paper.

(3) Determination of signal onsets for time differences : This is not explained clearly enough and it would be nice to provide more details. Define what "slope" is! Did

you use the raw waveforms or the signal envelope? Any references for this method? Is it similar to STA/LTA? What is the time window around the pre-defined event used here? For such a low number of events, wouldn't manual picking be more precise and feasible? Is the onset time the same that is used to compute Veff? I am a bit surprised that cross-correlation does not work. Have you tried to use only the first, more coherent part of signal, not the whole event? I would expect that cross-correlation is a much more precise measure of time lags than any automatic picking algorithm.

Other comments:

page 1 line 12 : I would say the effect of calving is not just equal, but can also be larger than melting at individual glaciers.

page 2 line 2 : Calving seasonality in general is not only due to melange ice, but also due to increase in meltwater-induced sliding, ocean temperature variations, and ocean tides, etc ...

Page 2 line 16 – 21 : Other possible causes for seismic calving signals have been suggested (at least for calving styles observed in Alaska and Svalbard) : ice - sea-surface interactions (Bartholomaus et al., 2012, in JGR; Köhler et al., 2015, in Polar Research).

Page 2 line 30 : Another possibility of locating complicated signals without using pre-determined velocities of individual seismic phases are the use of small-aperture arrays. Directional information can be obtained by applying signal beamforming or Frequency-Wavenumber analysis which can than be triangulated. (Köhler et al, 2015, in Polar Research; Koubova, 2015: www.duo.uio.no/handle/10852/45791?show=full)

Data section : Are all calving events used here detected manually or is an automatic detector used? Are the calving events identified only based on inspection of frequency spectrum ? I would expect that regional earthquakes have energy above 1 Hz as well (see e.g. Köhler et al, Svalbard). Description of JIG station data is missing.

Page 3 line 15 : "teleseismic events from regional earthquakes" : Please rephrase : Either it is a teleseismic earthquake or a regional earthquake. In seismology there is a clear distinction between both events.

Page 6 Line 18 : What do you mean with "only two seismometers". Is the signal too noisy on the others stations? If not, why not use all stations for a robust estimation of veff? What are the individual measurements for all stations and all calving events? Is there really no difference between Hellheim and Jakobshavn?

Fig 5 : Two hyperbolas are shown for each velocity. I suppose they correspond to two stations pairs. Which of the three possible station pairs to they correspond to? Why not plot all three hyperbolas? Also, indicating the exact location of calving front at the time of the calving event would be helpful. Then one could see how an individual hyperbola intersects with the terminus. After all, this is what the authors suggest: a two-station method. It looks like Veff=1.4 could be as good, or even closer to the front. Furthermore, the authors write that the calving events appears to be in the ocean. However, it is actually located on the glacier (the melange is in the west, isn't it?).

Page 7 line 3 : "teleseism", write teleseismic earthquakes, see my comment about regional earthquakes above

Discussion about velocity: What about the more distant station at Jakobshavn (JIG 1 and 2)? The signal would have to travel through a lot of rock (possibly at sea bottom) I guess.

Discussion about depth: I am not sure if the main limitation for depth resolution is the missing velocity model for the glacier. One simply needs more stations close and above the source (on-ice) for a more precise and accurate localization. Also, I actually don't see the need the determine the depth of calving. Calving is usually affecting the whole height (or a big part) of the terminus (except maybe submarine events or small pieces of ice). However, I agree that depth may be relevant to analyze precursor events like fracturing.

[Figure]

Page 9 line 2: Even for a station at same elevation, P waves could come from below (refracted, diving waves).

Brune model: I am not sure if this source model can be applied here. If calving signals are associate with a simple rupture process I would agree. However, many mechanisms have been suggested (ice-sea-surface interactions, interaction with fjord bottom, forces that cause change in the motion of the ice after and during calving (glacial earthquakes at Hellheim, Murray et al, 2015)). I doubt that it is mainly the rupture signal that we see on the seismometers ...

Fig. 8 : Can you indicate the front retreat on the map? Is it consistent with the event locations?

In Conclusion: "...get around the emergent P-wave problem" : I don't agree. You still have to deal with the emergent onset, i.e. to pick an arrival to determine the time lag (see comment above). That, and estimating the velocity, are basically the same tasks for traditional travel-time based localization methods.

---

## Referee Comment (RC2) · J. Amundson (Referee) · 16 Jun 2016

This paper presents a method for detecting and locating calving events using local seismic observations. Localization of calving events has been challenging due to the emergent nature of seismograms produced by calving icebergs. Therefore I think that this paper will be an important contribution to the growing field of glacier seismology.

For the most part the paper is clearly written and easy to follow. However I do have some questions regarding the calculation of the time delay and the limitations of the method, and in some cases I think the paper could be expanded to address glaciological questions in more detail.

- When you attempted to cross-correlate the seismograms, did you use the entire seismogram (e.g., similar to what is shown in Figure 2) or just some subset of the seismo-

gram? If the former, one issue that might cause problems is that calving events often involve the detachment of multiple icebergs. I would guess that the calving events shown in Figure 2 involved 2–3 icebergs. If the latter, how did you determine what subset to pick? Also perhaps show what section of data you are using, or refer to Figure 12 when describing the section of data that you used. Its not clear exactly what data you are using.

- I understand that cross-correlating the seismograms did not give satisfactory results for the calculating the time delay. Did you also consider cross-correlating the envelope of the waveforms? I'm wondering if there is a more robust way of calculating the time constant that doesn't rely on an empirical constant.

- This studied focused on large, full-thickness calving events that occur on weekly timescales. These are events that, at least for focused studies on individual glaciers, can often be located using time-lapse or satellite imagery (as stated in the paper). Smaller calving events clearly occur more frequently. Admittedly, these smaller events may be insignificant for the total mass loss from glaciers like Helheim and Jakobshavn, but understanding the variability of these smaller events may provide insights into processes driving calving. If you decrease the STA/LTA threshold, can you detect and locate more events? If so, what sort of patterns emerge?

- Another way to expand the applicability of this method is to show that it works for regional seismic data. Full-thickness calving events at Jakobshavn Isbrae are detectable at ILULI (50 km away), SFJD (250 km away) and sometimes SUMG (400? km away), even when the calving events don't generate classic "glacial earthquakes". Have you tried incorporating regional seismic data into your method? Can a regional seismic network such as GLISN be used to detect and locate large calving events around Greenland (besides those that generate glacial earthquakes) using your methodology?

A few more minor questions:

- The authors state that calving at Helheim preferentially occurs on the north side of the

glacier. Is this where the glacier is thickest/fastest? Is this statement really just saying that full-thickness calving events only occur in that region?

- What is the date on the googleearth imagery used in the figures? Perhaps it makes sense to use newer imagery that was captured closer to the time of study (e.g., from Landsat 8)?

- I would guess that two or three icebergs calved during the events that appear in Fig. 2. What happens if you analyze each of the peaks in seismicity separately? Do you see calving propagating upglacier or across the glacier face?

- page 9, equation for radius of a circular fault: the equation contains $\beta_0$, but the text describing the variables only refers to $\beta$. Should these be the same thing?
* * *

---

## Referee Comment (RC3) · A. Diez (Referee) · 29 Jun 2016

Mei et al. analyze passive source seismic data mainly from Helheim glacier to localize calving events. For the localization they pick the first arrival of the seismic signal of the calving event. Combined with a predetermined velocity hyperbolas are calculated to determine the source location. This method is used for calving events at Helheim between Sep 2014 and Jul 2015, localizing 11 events in total. Finally, the authors use these events to determine the size of the calving event and speculate that the clustering of the calving events on the northern half of Helheim might be due to larger ice thickness and differences in surface roughness. The paper uses a seismic method not applied for the localization of calving events before. It is great to see a different method applied to the subject of localizing calving events from nearby seismometers. The paper is in most parts easy to understand. The method should be explained in a

bit more detail in certain parts and I do have some questions regarding the validity on how the method is applied here.

General comments:

It is not a hundred present clear to me, what the main focus of the paper is. Is it to introduce the hyperbolic method for the localization of calving events and Helheim glacier is just an example of the application of this method, or is it the localization and interpretation of the calving events for which the hyperbolic method is introduced? I think that should be clarified and the text adjusted accordingly.

You use a lot of fill words and subjective descriptions, that make sentences unnecessarily long (also, some, severely, powerful). Readability and understandability would increase significantly if the sentences were shorter and the sentence structure less complex. Often it would be easy to split one sentence into two sentences.

Chapter 3 Hyperbolic Method: I do have some question regarding the method: - Why does the cross correlation not work? Are the waveforms so different due to the difference in interference of the different wave types at the different stations? Could you please clarify this? Did you try different bandpass filters and window length for the cross correlation.

-How big is the error when you pick the first arrival (estimate), what does this mean for the precision of your localization?

- If you do not determine the wave type how can you be sure that the first break you are picking is coherent. Most likely and in most cases you will pick the surface wave. Which would be totally valid, and you later state that it is the surface wave you are analyzing. So why not determine the phase you are using for the analysis and use surface waves. My fear with this technique is that you might have a seismometer close to the source and it is not possible to see the P-wave first arrival, so you would pick the surface wave. For a seismometer that is further away the P-wave and surface wave

might be separated better, hence the wave you pick would be the P-wave. But if you pick different wave phases at different stations how do you want to use one velocity to find the correct location of your source. Imagine you pick the P-wave at seismometer 1 and the surface wave at seismometer 3. For the analysis you then use the velocity of 1.17 km/s, your localization would be totally wrong. This is a crucial point and the way I understand your analysis I can't see that the analysis is correct as you apply it. Please clarify this!

- How was the location of the calving events observed by persons determined. Where this events filmed? Small errors in the location of the observed calving events will lead to big errors in the derived velocity. How do you derive such a small error as 0.1 km/s? Please clarify how this velocity is determined in more detail.

- How can you use the data from Jacobshavn to determine the velocity. It's a completely different setting then Helheim. At Helheim your seismometers are located inland of the glacier front, i.e., waves will travel a large part through glacier ice. At Jacobshavn the seismometer are locates, mostly (except of seismometer 3), downstream of the glacier front, i.e. waves mainly travel through water and ice mélange. You must derive totally different velocities for these two locations.

- Did you try a grid search. As you do have multiple seismometers you could use the derived lag of all combination and find the global maximum testing different directions and velocities.

Discusssion: Large parts of the Discussion are not a discussion but an interpretation of the results or even speculation of what their causes are. This needs to be clearly differentiated, discussion and interpretation.

Determination of magnitude: For the method of Brune, you say, you have to use the corner frequency of the S-wave. But you don't use the S-wave, so why should that method be valid here at all. Further, I have trouble seeing the corner frequency between 1-5 Hz in your plots in Fig. 12. And why do you choose this small time interval

you are using for the calculation of this spectrum?

Figures: Must appear in the order in which they appear in the Text. Fig 6 – page 6 line 6, Fig 5 – page 6 line 20. Always refer to the Figure by number, not see the above Figure. It is not necessary to write (see Figure . . .) instead (Figure . . .) is sufficient. It is totally clear that I'm supposed to have a look at the Figure.

Considering merging Fig 1 and Fig 2. One subplot of these two Figures will be enough to show the difference.

Google earth figures: I think it would be more appropriate to use maps or satellite images like Landsat here (http://earthexplorer.usgs.gov/). Further these images need, some reference frame, coordinates, a north arrow, a map where we are in Greenland.

Figure 1: Why don't you use the transfer function of the seismometers to show the data as displacement? That will be much easier to understand for someone not that familiar with passive seismic data.

Figure 11: I don't think that Figure is necessary. It can be well seen on Figure 8.

Specific comments:

For line specific comments see the attached PDF.

Please also note the supplement to this comment:
http://www.the-cryosphere-discuss.net/tc-2016-85/tc-2016-85-RC3-supplement.pdf

[Figure]

**Supplement:**

[revised manuscript text omitted]

---

## Author Comment (AC1) · 2 Jul 2016

**We would like to thank the reviewer for their helpful and detailed review. Our responses are bolded.**

First of all, I would to thank the authors for their contribution to the progress in the emergent field of cryo-seismology. Studies like this can help to improve and establish passive seismic methods for monitoring and better understanding glacier dynamics. (1) My main comment is that the method is actually not new in seismology or acoustics, but has been suggested as a simple graphical signal localization method before, e.g. in:
J. Pujol (2004)
[...]
I agree that this study present the first application of this method to calving localization. However, this is not such a different approach compared to using traditional localization methods based on first onsets (without using S waves). Basic processing and assumptions are the same: you have to know the velocity model and to pick first arrival. Just using P wave onsets, it is necessary to use some constrains on source locations, but you do the same here when choosing one hyperbola.
The authors should clearly write this and state some references.

**Thank you for pointing this out! We have now added the original hyperbolic method paper (Mohorovicic 1915) to the introduction and referenced Pujol (2004) who suggests that this method is best for shallow events where refraction along a bottom interface is insignificant.**

(2) The title suggests a two-station method. This is true if the location of the terminus is known. However, in the paper the authors only present results of using more than two stations at Hellheim and Jakobshavn glacier (hyperbola intersection). So I would suggest to change the title or put more emphasize on two-station results in the paper.

**We intended "two" to mean "pair-wise", i.e. you need two stations to define each hyperbola. We have changed the title to "Calving Localization at Helheim Glacier Using Multiple Seismic Stations" for clarity and added "paired" to the abstract description.**

(3) Determination of signal onsets for time differences : This is not explained clearly enough and it would be nice to provide more details. Define what "slope" is! Did you use the raw waveforms or the signal envelope? Any references for this method? Is it similar to STA/LTA? What is the time window around the pre-defined event used here? For such a low number of events, wouldn't manual picking be more precise and feasible? Is the onset time the same that is used to compute Veff ? I am a bit surprised that cross-correlation does not work. Have you tried to use only the first, more coherent part of signal, not the whole event? I would expect that cross-correlation is a much more precise measure of time lags than any automatic picking algorithm.
**Added details: the slope is from the raw waveforms, and this method is entirely empirical. We were unwilling to do a manual method because this would be subjective, but our automated method still requires manual checking (and our 44% value was manually/empirically determined). The time window for cross-correlation was the 25s window pictured in Fig. 4. Cross-correlating the entire event doesn't work either, we suspect this is because the same calving event can look very different at two different**

**stations:**

[Figure]

Fig. A1. The same calving event at two different stations.

**These signals have such different shapes that cross-correlation does not work. It would probably work better with just the wave envelope, but to create that, there would probably be some rounding or other shift of at least 0.5s, which is significant as most of our lags are <3 s. It may be better just to state that we are manually detecting the events and using our automated 44% trigger to know what neighborhood to inspect.**

Other comments:
page 1 line 12 : I would say the effect of calving is not just equal, but can also be larger than melting at individual glaciers.
**Noted and changed.**
page 2 line 2 : Calving seasonality in general is not only due to melange ice, but also due to increase in meltwater-induced sliding, ocean temperature variations, and ocean tides, etc …
**Noted and changed.**
Page 2 line 16 – 21 : Other possible causes for seismic calving signals have been suggested (at least for calving styles observed in Alaska and Svalbard) : ice - sea-surface interactions (Bartholomaus et al., 2012, in JGR; Köhler et al., 2015, in Polar Research).
**Noted and changed.**

Page 2 line 30 : Another possibility of locating complicated signals without using pre-determined velocities of individual seismic phases are the use of small-aperture arrays. Directional information can be obtained by applying signal beamforming or Frequency-Wavenumber analysis which can than be triangulated. (Köhler et al, 2015, in Polar Research; Koubova, 2015: www.duo.uio.no/handle/10852/45791?show=full)
**Based on my reading of Köhler 2015, it seems like the method is manual phase-picking of P/S waves to generate a backazimuth and distance, as their array is regional and thus far enough to distinguish different phases? We have added Koubova's description of beamforming, though it seems to rely on having a backazimuth already (which we do not have in our case until after the location is determined).**

Data section : Are all calving events used here detected manually or is an automatic detector used? Are the calving events identified only based on

inspection of frequency spectrum ? I would expect that regional earthquakes have energy above 1 Hz as well (see e.g. Köhler et al, Svalbard). Description of JIG station data is missing.
**The JIG data are now removed from the study as we should not reasonably expect that surface velocities are equivalent to Helheim. Events are detected semi-automatically using a Signal/Noise threshold then individually inspected to rule out regional earthquakes, which have different frequency breakdowns – see Figure A2.**

[Figure]

(a) Calving event          (b) Regional earthquake

**Fig. A2. The spectra (top) and spectrograms (bottom) for a calving event (a) and a regional earthquake in Iceland 960km away (b).**

Page 3 line 15 : "teleseismic events from regional earthquakes" : Please rephrase : Either it is a teleseismic earthquake or a regional earthquake. In seismology there is a clear distinction between both events.
**Thanks, noted. The earthquake is ~960km from the station which is very near the teleseismic threshold, but we have gone with just "regional" instead.**

Page 6 Line 18 : What do you mean with "only two seismometers". Is the signal too noisy on the others stations? If not, why not use all stations for a robust estimation of veff? What are the individual measurements for all stations and all calving events? Is there really no difference between Hellheim and Jakobshavn?
**"Only two seismometers" is because only two seismometers were deployed in August 2013-2014, and the other two seismometers were deployed after the calving event had occurred. What do you mean by "individual measurments of all stations"? The other calving events that form our catalog were not observed (though they were confirmed to be calving events using MODIS Terra satellite imagery). The Jakobshavn event has been removed because even if the velocities seem similar, it may be coincidental and we should not give any weight to the Jakobshavn data.**

Fig 5 : Two hyperbolas are shown for each velocity. I suppose they correspond to two stations pairs. Which of the three possible station pairs to they correspond to? Why not plot all three hyperbolas? Also, indicating the exact location of calving front at the time of the calving event would be helpful. Then one could see how an individual hyperbola intersects with the terminus. After all, this is what the authors

suggest: a two-station method. It looks like Veff=1.4 could be as good, or even closer to the front.

Furthermore, the authors write that the calving events appears to be in the ocean. However, it is actually located on the glacier (the melange is in the west, isn't it?).
**Yes – the melange is actually west. Though, this plot has been removed in any case (we are removing all references to Jakobshavn).**

Page 7 line 3 : "teleseism", write teleseismic earthquakes, see my comment about regional earthquakes above
**Noted, thanks**

Discussion about velocity: What about the more distant station at Jakobshavn (JIG 1 and 2)? The signal would have to travel through a lot of rock (possibly at sea bottom) I guess.
**Removed Jakobshavn event.**

Discussion about depth: I am not sure if the main limitation for depth resolution is the missing velocity model for the glacier. One simply needs more stations close and above the source (on-ice) for a more precise and accurate localization. Also, I actually don't see the need the determine the depth of calving. Calving is usually affecting the whole height (or a big part) of the terminus (except maybe submarine events or small pieces of ice). However, I agree that depth may be relevant to analyze precursor events like fracturing.
**Our reviewer #3 pointed out that depth is not a well-posed question – for an event that removes an entire column of ice, there is no real 'depth' (unless you use the depth of the entire glacier, which is measurable with bathymetry). We will minimize the discussion of depth.**

Page 9 line 2: Even for a station at same elevation, P waves could come from below (refracted, diving waves).
**True. For our study, we show that the wave arrivals are dominated by surface (Rayleigh) waves, and so we are able to neglect refracted/diving waves.**

Brune model: I am not sure if this source model can be applied here. If calving signals are associate with a simple rupture process I would agree. However, many mechanisms have been suggested (ice-sea-surface interactions, interaction with fjord bottom, forces that cause change in the motion of the ice after and during calving (glacial earth- quakes at Hellheim, Murray et al, 2015)). I doubt that it is mainly the rupture signal that we see on the seismometers …
**Agreed – the Brune model was intended as a comparison to see if it were a rupture, what size it would be. Do you think we should not mention Brune at all, or qualify it more (that it may not be a rupture signal at all, but if it is, then it has size 50m)?**

Fig. 8 : Can you indicate the front retreat on the map? Is it consistent with the event locations?
**Updated figure. Figure A3 shows that the calving fronts are close to the events – a good check to make!**

[Figure]

**Figure A3. Catalogue of eleven calving events localized on Helheim glacier, showing the movement of the calving front for certain dates (taken from Landsat 8).**

In Conclusion: "…get around the emergent P-wave problem" : I don't agree. You still have to deal with the emergent onset, i.e. to pick an arrival to determine the time lag (see comment above). That, and estimating the velocity, are basically the same tasks for traditional travel-time based localization methods

**A fair point – we have changed this to "We find that the local seismic signals are dominated by surface (Rayleigh) waves, which makes distinguishing between different seismic wave components (a key benefit of regional arrays) irrelevant. A local array is able to localize calving with greater resolution than a regional array. Identifying the signal onsets is not fully automated and requires manual inspection of signals, due to the emergent signals involved in glacial calving."**

---

## Short Comment (SC1) · 5 Jul 2016

Hi Anja,

Thanks for your very detailed review. I'm implementing the grid search now - one issue is that each event returns a different [x, y, velocity] corresponding to the global minimum (I am using minimum in total different in reported and gridded lags). That is, for my 11 events, I may end up with 11 different surface velocities between say 1-1.4 km/s - but there should be the same velocity for all the events.

What do you suggest?

I am thinking taking the average of all the optimal velocities then redoing the grid search in 2D for just [x, y] given this velocity value?

---

## Short Comment (SC2) · 5 Jul 2016

The gridsearch gives similar results to the hyperbolas. Only two of the events have grid search critical points outside of the hyperbolas in our method (though, as per my previous response, if we set a particular v_surface then this may go away). For both the points, if I plot the top 100 results instead of just the top 1 (given my gridsearch has ~7million points) result then they are inside the hyperbolas.

[Figure]

**Fig. 1.** Calving events with 'x' indicating the gridsearch result and the hyperbolas as before.

---

## Author Comment (AC2) · 5 Jul 2016

Thanks for your helpful comments! Our responses are **bolded**.

- When you attempted to cross-correlate the seismograms, did you use the entire seismogram (e.g., similar to what is shown in Figure 2) or just some subset of the seismogram? If the former, one issue that might cause problems is that calving events often involve the detachment of multiple icebergs. I would guess that the calving events shown in Figure 2 involved 2–3 icebergs. If the latter, how did you determine what subset to pick? Also perhaps show what section of data you are using, or refer to Figure 12 when describing the section of data that you used. Its not clear exactly what data you are using. **We cross-correlated the snippets like those in Fig. 12, which were manually chosen by looking at sharp peaks in the spectra. We required that all four stations had clear peaks (so that we could identify lags) – in some cases this was not possible. We have added text to clarify that we are using the windows in Fig. 12 in the cross-correlation.**

- I understand that cross-correlating the seismograms did not give satisfactory results for the calculating the time delay. Did you also consider cross-correlating the envelope of the waveforms? I'm wondering if there is a more robust way of calculating the time constant that doesn't rely on an empirical constant. **Our reviewer #1 suggested just to manually pick out the lags due to the small sample size. We were not wanting to do it fully manually due to the inherent subjectivity. Cross-correlating the envelopes would lose the resolution of the lags as the envelope is of order 5 s (in Fig. 6) but we have lags of order 1-3 s or so and even a 0.5 s shift would dramatically change the hyperbola. We were unable to think of a truly automated way of calculating the lags – we could also do a STA/LTA method but all methods require manual checking to see if the results are sensible (like for cross-correlation, we found implausible lags). Do you think it would be better just to concede that even our most automated method requires manual verification, and to just scrap our emperical values and do the localization manually?**

- This study focused on large, full-thickness calving events that occur on weekly timescales. These are events that, at least for focused studies on individual glaciers, can often be located using time-lapse or satellite imagery (as stated in the paper). Smaller calving events clearly occur more frequently. Admittedly, these smaller events may be insignificant for the total mass loss from glaciers like Helheim and Jakobshavn, but understanding the variability of these smaller events may provide insights into processes driving calving. If you decrease

the STA/LTA threshold, can you detect and locate more events? If so, what sort of patterns emerge?

**The biggest issue here is that the lower-amplitude events have a smaller slope (because they have a lower amplitude change in more-or-less the same time time step) and so our automated lag detector is much less accurate. Moreover, as the peaks are less sharp, it's also harder to manually identify them. However, if there is relatively low noise around the signal, and there is a short, sharp burst (like at 03:06 in Fig A3 below), then our method does converge. But because calving signals are emergent, we don't have these short sharp bursts during the main calving event. So we are unlikely to be able to localize the main calving signal for small events, but we may be able to localize nearby secondary signals from these events – though this probably would not localize the calving event itself.**

- Another way to expand the applicability of this method is to show that it works for regional seismic data. Full-thickness calving events at Jakobshavn Isbrae are detectable at ILULI (50 km away), SFJD (250 km away) and sometimes SUMG (400? km away), even when the calving events don't generate classic "glacial earthquakes". Have you tried incorporating regional seismic data into your method? Can a regional seismic network such as GLISN be used to detect and locate large calving events around Greenland (besides those that generate glacial earthquakes) using your methodology?

**Kira Olsen presented a poster at AGU 2015 that looked at locating glacial earthquakes using GLISN (using an azimuthal method), though due to the distance of the seismometers, this localization was limited to identify which glacier calved and not where on each glacier the calving occurred. We do not believe our method easily scales up to regional arrays, as a key assumption (that the surface velocity is constant in all directions) is no longer valid with different travel paths from the epicenter – this means the locus would no longer be a hyperbola, but rather some (more complex) other shape. Also, at higher distances, the error in v_eff would make the localization too imprecise (likely over most of the glacier). Though, the main limitation is the lack of a constant v_eff. Another test for the validity of our method is to look at calving fronts and to see if our localization matches – and our method does hold (Figure A2 below).**

A few more minor questions:
- The authors state that calving at Helheim preferentially occurs on the north side of the glacier. Is this where the glacier is thickest/fastest? Is this statement really just saying that full-thickness calving events only occur in that region?

**This is where the glacier is thickest (Fig A2), by about 200 m.**

[Figure]

Fig A1. Topography of the rock below Helheim Glacier, taken from NSIDC McORDS flights collated between 2008-2012.

- What is the date on the googleearth imagery used in the figures? Perhaps it makes sense to use newer imagery that was captured closer to the time of study (e.g., from Landsat 8)?

**Yes – we have now updated this (Fig A2) to use a Landsat image from July 2015, and we use three other images to also show the progression of the calving front (at the suggestion of reviewer #1).**

[Figure]

**Fig. A2. Updated calving catalog showing locations and the movement of the calving front.**

- I would guess that two or three icebergs calved during the events that appear in Fig. 2. What happens if you analyze each of the peaks in seismicity separately? Do you see calving propagating upglacier or across the glacier face?

**The calving event Fig. 2 is from August 2014, when only two seismometers were available, so unfortunately we are unable to localize those events. However, we found another multiple-calving event (June 6 2015) which we treated the peaks separately (Fig. A3). It looks like the calving progression in that figure goes Red, Yellow, then it splits in two directions and goes Blue-Purple and Green. The estimated calving localizations all touch the calving front of June 5 (from Landsat), which is good. Do you think this plot is worth adding to the manuscript? This event is part of the calving catalogue from the manuscript you already saw, but only the "main" event (i.e. the highest peak at 2:30) is used. The plot may be a bit misleading because the shapes may suggest calving magnitude, which is not the case here (the biggest area in purple actually represents the smallest amplitude signal).**

[Figure]

Fig A3. Multiple-iceberg calving event on June 6 2015.

- page 9, equation for radius of a circular fault: the equation contains beta_0, but the text describing the variables only refers to beta. Should these be the same thing?

**Yes – thanks.**

---

## Referee Comment (RC4) · Anonymous Referee #1 · 11 Jul 2016

>Based on my reading of Köhler 2015, it seems like the method is manual phase-picking of P/S waves to generate a backazimuth and distance, as their array is regional and thus far enough to distinguish different phases? We have added Koubova's description of beamforming, though it seems to rely on having a backazimuth already (which we do not have in our case until after the location is determined).

We (revealing my identity here) used an array at near-regional distance to pick P and S waves and to measure the backazimuth. That was done both manually (for the largest events), but also automatically. However, my point here was just that you should mention the alternative to locate complex seismic signals (such as calving) with local small-aperture arrays. If you have 2 or 3 of these arrays close to the terminus, azimuth intersection allows to locate the events without having to identify / pick individual seismic phases. Of course application of this method is limited by instrument availability.

[Figure]

>Agreed – the Brune model was intended as a comparison to see if it were a rupture, what size it would be. Do you think we should not mention Brune at all, or qualify it more (that it may not be a rupture signal at all, but if it is, then it has size 50m)?

Since there is still some ongoing discussion about the source mechanisms of calving events, you may keep it. However, in this case you should mention and briefly discuss the other source candidates as well.

---

## Author Comment (AC3) · 30 Jul 2016

Mei et al. analyze passive source seismic data mainly from Helheim glacier to localize calving events. For the localization they pick the first arrival of the seismic signal of the calving event. Combined with a predetermined velocity hyperbolas are calculated to determine the source location. This method is used for calving events at Helheim between Sep 2014 and Jul 2015, localizing 11 events in total. Finally, the authors use these events to determine the size of the calving event and speculate that the clustering of the calving events on the northern half of Helheim might be due to larger ice thickness and differences in surface roughness. The paper uses a seismic method not applied for the localization of calving events before. It is great to see a different method applied to the subject of localizing calving events from nearby seismometers.

The paper is in most parts easy to understand. The method should be explained in a bit more detail in certain parts and I do have some questions regarding the validity on how the method is applied here.

**Thank you for your extremely detailed review. Our responses are in bold.**

General comments:
It is not a hundred present clear to me, what the main focus of the paper is. Is it to introduce the hyperbolic method for the localization of calving events and Helheim glacier is just an example of the application of this method, or is it the localization and interpretation of the calving events for which the hyperbolic method is introduced? I think that should be clarified and the text adjusted accordingly.
**After we learned from review 1 that the hyperbolic technique is already used in acoustics/seismology (but not yet for glacial calving), we have shifted the focus of the paper to the localization and interpretation of calving events for which we use the hyperbolic method (and also a grid search method as you suggest). Our revision of the manuscript hopefully reflects this better.**

You use a lot of fill words and subjective descriptions, that make sentences unnecessarily long (also, some, severely, powerful). Readability and understandability would increase significantly if the sentences were shorter and the sentence structure less complex. Often it would be easy to split one sentence into two sentences.
**Thanks. We have gone through the manuscript to try and break up long sentences.**

Chapter 3 Hyperbolic Method: I do have some question regarding the method: - Why does the cross correlation not work? Are the waveforms so different due to the difference in interference of the different wave types at the different stations? Could you please clarify this? Did you try different bandpass filters and window length for the cross correlation.
**We believe cross-correlation does not work because the signals sometimes look very different in different directions. This may be due to the orientation of the fracture. There is no clear pattern to which pairings do not have successful cross-correlation. Possibly this is also because when the calving event happens, the icebergs are formed on the eastern side, and this is non-symmetric to the western side so the signal is not radially identical. It is also possible, as you state, that there is interference from linearly polarized waves (e.g. the P-wave) that affect the shape of the waveforms at each station differently. We tried different filters both for high and low and both for the overall waveform shape and for the small peaks, but we could not get good values for all events and all seismometer pairings, and as a result we could not use this fully automated method.**

-How big is the error when you pick the first arrival (estimate), what does this mean for the precision of your localization?
**So the estimate of the lag (i.e. a subtraction of the arrivals) should have any systematic error removed by the subtraction. One reason we did not want to pick out arrivals by**

**eye is because of the error that this would create. The random error of the actual estimation is hard to quantify as the true arrival is not known. Signal onset detection is still an ongoing area of research (e.g. Ross & Ben-Gurion 2014). One way to quantify the error is to use the error of localization (compared to camera-observed events) and then reverse-engineer the time lag.**

**Ross, Z. E., & Ben-Zion, Y. (2014). Automatic picking of direct P, S seismic phases and fault zone head waves. *Geophysical Journal International, 199*(1), 368-381.**

- If you do not determine the wave type how can you be sure that the first break you are picking is coherent. Most likely and in most cases you will pick the surface wave. Which would be totally valid, and you later state that it is the surface wave you are analyzing. So why not determine the phase you are using for the analysis and use surface waves. My fear with this technique is that you might have a seismometer close to the source and it is not possible to see the P-wave first arrival, so you would pick the surface wave. For a seismometer that is further away the P-wave and surface wave might be separated better, hence the wave you pick would be the P-wave. But if you pick different wave phases at different stations how do you want to use one velocity to find the correct location of your source. Imagine you pick the P-wave at seismometer 1 and the surface wave at seismometer 3. For the analysis you then use the velocity of 1.17 km/s, your localization would be totally wrong. This is a crucial point and the way I understand your analysis I can't see that the analysis is correct as you apply it. Please clarify this!
**This is a good criticism. For our method, we checked all the particle plots for each event to conclude that they are all dominated by surface (Rayleigh) waves. We originally thought it fit better in the Discussion of what waves were being detected, but this analysis was done before making the localization catalog. It is important as a premise for our technique so we have moved this section to the Methods.**

- How was the location of the calving events observed by persons determined. Where this events filmed? Small errors in the location of the observed calving events will lead to big errors in the derived velocity. How do you derive such a small error as 0.1 km/s? Please clarify how this velocity is determined in more detail.
**The events were filmed, but you are right there could be small errors in the viewed localization, so we are changing our velocity estimation method. We used a grid search as you later suggest, and got a best fit of 1.20 +- 0.03 km/s. We have updated our plots (this changes the locations only slightly). 0.1 is the standard deviation of the best fit velocities for our 11 events, and so the standard error for these 11 points would be 0.1/sqrt(11) = 0.03 km/s.**

- How can you use the data from Jacobshavn to determine the velocity. It's a completely different setting then Helheim. At Helheim your seismometers are located inland of the glacier front, i.e., waves will travel a large part through glacier ice. At Jacobshavn the seismometer are locates, mostly (except of seismometer 3), downstream of the glacier front, i.e. waves mainly travel through water and ice mélange. You must derive totally different velocities for these two locations.
**Yes, you are right. We have deleted this section and determined our velocity using grid search with Helheim only.**

- Did you try a grid search. As you do have multiple seismometers you could use the derived lag of all combination and find the global maximum testing different directions and velocities.
**Thanks for this suggestion. We applied a grid search, as mentioned above, successfully.**

Discusssion: Large parts of the Discussion are not a discussion but an interpretation of the results or even speculation of what their causes are. This needs to be clearly differentiated, discussion and interpretation.
**We have now separated into an "interpretation of results" and a "discussion of methods" subsection.**

Determination of magnitude: For the method of Brune, you say, you have to use the corner frequency of the S-wave. But you don't use the S-wave, so why should that method be valid here at all. Further, I have trouble seeing the corner frequency between 1-5 Hz in your plots in Fig. 12. And why do you choose this small time interval you are using for the calculation of this spectrum?
**This method is intended as a comparison of traditional seismic techniques to see what a fracture size would be following the Brune model. The S-wave velocity is needed for this, and we do not have S-waves as you mention, but we use the relationship between S-wave speed and surface wave speed (as a function of the Poisson ratio) to estimate the S-wave velocity. The small time interval is because we believe the high-amplitude peak corresponds to the calving event, and so we want to estimate the fracture size from this particular window.**

Figures: Must appear in the order in which they appear in the Text. Fig 6 – page 6 line 6, Fig 5 – page 6 line 20. Always refer to the Figure by number, not see the above Figure. It is not necessary to write (see Figure...) instead (Figure...) is sufficient. It is totally clear that I'm supposed to have a look at the Figure.
**Thanks for this, we have checked the order carefully and removed "see" from whenever we mention figures.**

Considering merging Fig 1 and Fig 2. One subplot of these two Figures will be enough to show the difference.
**Yes, we have now done so.**

Google earth figures: I think it would be more appropriate to use maps or satellite images like Landsat here (http://earthexplorer.usgs.gov/). Further these images need, some reference frame, coordinates, a north arrow, a map where we are in Greenland.
**Thanks for this. We have now switched to Landsat images with a reference frame, grid, north arrow and a map.**

Figure 1: Why don't you use the transfer function of the seismometers to show the data as displacement? That will be much easier to understand for someone not that familiar with passive seismic data.
**We think it is important to show the different phases of the calving event (to highlight its emergent nature), as well as to show the similarities of these signals to Amundson 2010/2012 etc. We have explained a bit more clearly what the trace is showing.**

Figure 11: I don't think that Figure is necessary. It can be well seen on Figure 8.
**Ok. We have removed this.**

Specific comments:
For line specific comments see the attached PDF.
**Thanks for your line comments too. We have adapted most of your suggestions. We have chosen to keep "emergent" as a description as this is commonly used to describe calving events (e.g. Amundson 2012, Richardson 2010) though we have now added a clearer description of what "emergent" means.**

---

## Author Response (AR2)

The authors have addressed my comments from my first review. I now have only a few minor comments for the authors to address.
**Thanks for your second review. Responses are in bold.**

Line 3: It kind of feels like there is a sentence or phrase missing here. Perhaps: "The crossing points of the hyperbolas provide an estimate of the location of the calving event." If acceptable, then delete the sentence starting with "Using local stations..."
**Change made.**
Line 4: Suggest "emergent nature of calving-generated seismograms". You mean that the seismograms are emergent, and not the calving events themselves (correct?).
**Added "signals" after "calving".**

Line 15: Although this statement is correct, what is especially unique about tidewater glaciers is that they can undergo very large and rapid changes in volume with minimal climate forcing and that their sensitivity to climate depends on where their termini are located.
**Changed to "Calving glaciers can rapidly advance and retreat in response to minimal climate signals, which can rapidly change the sea level"**

Line 5: "variations in meltwater sliding"? Do you mean "variations in basal motion due to meltwater input"?
**Change made**

Line 8: Suggest emphasizing that you are referring to terrestrial radar.
**Added "ground-based"**
Line 12: Delete "(providing seismic data)"
**OK.**

Lines 19-20: This sentence seems out of place and unnecessary.
**Changed to "One way to monitor glaciers and detect calving is to use seismic arrays". The purpose of this sentence is to introduce what the STA/LTA data comes from.**
Lines 21-22: Tsai et al proposed that glacial earthquakes were caused by icebergs pushing off of the glacier terminus.
**Yes – what he calls 'non-equilbrium calving' - so we delete "large" from the start of the sentence.**

Line 2: Why is that intersection an estimate of the size of the calving event, and not an indication of error in the method?
**This was misphrased on our part. It should read "their intersection to be an estimate of where the calving occured" - we did not intend to suggest the area was the calved area, but that the area was the area in which calving occurred.**

Line 5: "we have Figure 7" sounds awkward.
**Changed to "yields Fig. 7"**

Line 11: "regional" is misspelled.
**Thanks - fixed**

Line 29: Density varies very little with depth in glaciers and is generally treated as a constant. Crevasses obviously affect the bulk density, and air bubbles in the ice have a small impact on the density. It seems unlikely to me that seismic wave velocities would change much as a result of variations in ice density.
**Ok – removed this sentence.**

Line 11: Suggest "as the calving events occur" (delete "are")
**fixed.**

The revised manuscript has improved and most of my previous comments have been addressed. However, I feel that some more revisions are necessary. There are some important issues the authors need to address and clarify, particularly, related to the gridsearch method and the use of camera imagery.

Introduction

The second paragraph needs to be reorganized a bit. The description of the motivation for calving monitoring, the direct calving monitoring methods available, and the use of seismic detection and subsequently localization methods is mixed and therefore a bit hard to follow. I would suggest the following structure:
(1) Motivation
For example, sentence from "There have not been enough .." (Page 2 line 2) until "... on a long-term basis to capture events." (line 6) should be moved after "The lack of understanding ..." on Page 1 line 18.
(2) Overview about direct calving monitoring methods and their disadvantages or limits: visual time lapse, satellite, radar, ...
(3) Seismic methods: Here, it would be good to clearly distinguish between event detection (for example STA/LTA) and localization (main focus of this study).
- Page 2 line 20: Rephrase sentence, for example to: " A common automated calving … is to use triggers based on short-time and long-time averages of the continuous seismic signal (STA/LTA)."
- Page 2 line 1: Sentence unclear. Rephrase maybe to: "To implement and validate detection and localization methods, visual confirmation of calving locations are required for example through terrestrial observations or (for sufficiently large events) satellite imagery."
**reworked the order of sentences, and also added " After an event has been detected, it can then be localized." to make it very clear that detection and localization are separate steps.**
Page 2 line 22: interaction with the sea surface
**fixed**
Page 2 line 16: Murray et al. 2015b: In that study basal crevassing is part of a model describing the calving of tabular icebergs. The seismic signal (glacier earthquake) is not directly related to the crevassing in that study or at least it is not discussed there. I think the mechanism described in Murray 2015a is more relevant here.
**Changed to "a mechanism for calving at Helheim" - we think the 2015b article is important as it focuses on Helheim calving, same as our study. 2015a is also mentioned earlier in the paragraph and we think both are worth mentioning in an introduction.**

Page 3 line 1: Rephrase, for example: "Another method to locate calving events, known as beamforming, uses the seismic signals recorded on several array stations to determine the time delay associated with a backazimuth that aligns the signals coherently (Koubova, 2015, Köhler et al. 2015)."
**changed.**

Data

Describe camera imagery data and indicate camera position in Fig 1.
The seismic detection method is not explained clearly, i.e., how is the catalog of 11 events obtained (STA/LTA? Visual inspection?). How complete is it? **Added to text. It is STA/LTA first then confirmed with visual from the local camera + satellite imagery, and it ignores smaller calving events that do not pass the STA/LTA cutoff (as these have less clear signal onsets).**

Hyperbolic method

Page 5 line 5: remove "describe the hyperbolic method and" and change to "... we apply the hyperbolic method ..."
**Ok, done.**

It might be helpful to mention that in order to draw a hyperbola, the parameter "b" needs to be determined, which is $b=\sqrt{c^2 - a^2}$ where 2c is the (known) distance between F1 and F2.
**Ok, done.**

The gridseach method is not explained sufficiently. It is just briefly described in the introduction, but the method section would be the proper place for it. I am not sure if I understand the processing flow completely. Please correct me and clarify in the text:

(1) Minimize residuum (dt – dt_measured) with dt = f(x,y,veff)=ds1/veff – ds2/veff, with ds1/2 being the distance between (x,y) and seismometer 1 and 2. If we would plot the residuum map over x/y, we would see that the error is minimal along hyperbolas. So the error is evaluated over all station pairs to obtain a distinct minimum: sum(dt – dt_measured). This is done for each calving event and the mean of all obtained velocities veff is computed. Is this correct?
(2) Draw hyperbolas using veff and get intersection area
(3) Repeat gridsearch with fixed veff

Why is step 3 done? Wouldn't it be more accurate to use directly the best location (x,y) found in the previous gridsearch? What is the benefit of drawing hyperbolas instead of simply using the gridsearch results? I am not sure if the intersection areas can be used as estimates for the calved areas (page 8 line 2). It seems to be more an estimate of the location uncertainty. Please clarify.
**The area is indeed an estimate of the uncertainty, as you previously mentioned our phrasing was unclear. The 'best location' using a variable (v) is not ideal for several reasons. In practical terms, having v as a variable means that you need 4 (stations instead of the usual 3 to triangulate. In more physical terms, we expect that the surface velocity should not depend on the calving event as the glacier surface should be more-or-less isotropic (at Helheim, this is not quite true – we talk about this in the discussion). Grid search (i.e. hyperparameter optimization) is a common statistical technique in machine learning which is why it's not described in greater detail, but we will add in a few more lines as it's a rather new in the earth sciences.**

Identifying signal lags

"1.44 standard deviations of all gradients": Please define what "all gradients" means. Do you mean all gradients measured within a time window around the event? How long is it?
Changed to "**1.44 standard deviations of all point-wise gradients at each time step of 0.025 s for the total time window in Figure 4"**

Page 7 line 4: The particle motion plots show Rayleigh waves, but also suggest a Love or SH wave contribution on the transverse component. So the conclusion that the signals are dominated by Rayleigh waves is too strong. I would rather say it is dominated by surface waves.
**OK, changed to 'surface'.**
Change "particle plots" to "particle motion plots" (also in Fig 5).
**OK, changed**

Page 7 Line 12: What is the reason for using these velocity limits (and not a wider range)?
**Just to save computational time – the hyperbolic method yielded an approximate velocity of 1.17 km/s so we took a neighborhood around this value. None of the optimized v_eff were below 1.05 km/s and we have an upper bound of around 1.4 km/s as a surface wave that**

**exceeds this velocity would not lead to hyperbolas intersecting on the glacier surface (it would be on the rock instead).**

Page 7 Line 13: This is the first time the "11 identified calving events" are mentioned. Are they identified from images or from the seismic data (visually or STA/LTA)? See comment in data section.
**Changed data section.**

Discussion

How complete is the calving catalog of 11 events? How does the ice loss corresponding to these events compare to the frontal retreat?
**Add a new plot to show the frontal retreat. We do not quantify the ice loss due to the imagery not being detailed enough.**

I would be very helpful to make more use of the images of the calving front which are mentioned several times. Maybe a few examples can be shown. Most important, can the images be used to estimate the locations and sizes of the 11 calving events? If so, how do these locations compare to the seismic locations? Maybe indicate these locations in Fig 7.
**The size is hard to estimate, but the locations are good to check against. This is in the new figure, as mentioned in previous comment.**

Brune model and fault size: As I wrote in my previous comments, I would be careful to interpret the resulting fault sizes too much. There might be some contribution of fracturing to the seismic signal, but faulting is probably not the dominant source mechanism here. I think it is worth to be discussed (as the authors do), but I believe it should be concluded more clearly that the dominant mechanism might for example be the one discussed by Murray et al. 2015a. As I stated earlier, in Murray et al 2015b basal crevassing is described as the initial part of the calving process, but the glacier earthquake generated is not directly related to crevassing. In this context, it would be interesting to investigate if there is an earlier part of the seismic calving signal that could be related to crevassing. Brune's source model could then be applied only to that part. Do you see anything like this in your calving signals?
**Ok, kept the discussion but emphasised that Murray 2015b is the more plausible mecahnism. We can't necessarily distinguish between what is the 'crevassing' part of the calving unless we also have (say) GPS measurements that tell us the procession of the calving event, so this will be a topic for future study.**

Page 12 line 18: Could this also be due to ice/glacier anisotropy?
**Yes, added.**

Page 13 line 3: Please explain what you mean with "not fully automatic"
**added "because it requires setting a gradient threshold manually, or manual checking the plausibility of cross-correlation results"**
page 13 line 6: remove "are" and "this mean that"
**removed "this mean that", could not find an extra "are".**

page 13 line 8: please explain "is not robust". What alternatives would you suggest to determine time lags? Please also mention the envelope cross-correlation (you did this in the review response, but this is also relevant in the manuscript).
**Changed 'robust' to "rigorous as it requires manual confirmation". The alternative is to use cross-correlations when it works and throw out examples when it doesn't work (which assumes there are a lot more events than the 11 we have). Added envelope stuff.**

[revised manuscript text omitted]